# Simulation Analysis of Sensors with Different Geometry Used in Measurements of Atmospheric Electricity [note 1]

**DOI:** 10.3390/s23249627

**Published:** 2023-12-05

**Authors:** Konrad Sobolewski, Marek Kubicki

**Affiliations:** 1Faculty of Electrical Engineering, Warsaw University of Technology, 00-661 Warszawa, Poland; 2Institute of Geophysics, Polish Academy of Sciences, 01-452 Warszawa, Poland; swider@igf.edu.pl

**Keywords:** atmospheric measurements, electric field sensors, modeling, simulations

## Abstract

The atmospheric electric current, “air–earth current”, flows between the low ionosphere and Earth’s surface. The source of this current is the potential difference between the global equalizing layer called the ionosphere and the ground surface. According to Wilson’s concept of the Earth’s Global Electric Circuit, in the areas of so-called fair weather, based on current measurements at the Earth’s surface, it is possible to conclude the global electrical processes in the ionosphere and higher layers. The theoretical basis for this inference is the law of continuity of electric current or the principle of conservation of electric charge. We present the results of simulations of the distribution of electric field lines for sensors with different geometries placed in a uniform electric field, representing the atmospheric electric field. The sensors are metal surfaces on which electric charges are induced or deposited. In the external measuring circuit to which the sensor is connected, an electric current [A] will flow, related to the air–earth current density [A/m^2^], but their relationship may be challenging to interpret. We analyze the impact of sensor geometry on the possibility of interpreting the atmospheric electric conduction and atmospheric displacement current based on the current measured in the external circuit. This present method can be used for the geometric construction of new sensors at the stage of determining the electrical characteristics of the sensor (e.g., effective collecting area). It can support the comprehensive design of a measurement system at the interface between an atmosphere, sensor, and electronic equipment.

## 1. Introduction

The electricity in the atmosphere in fair weather and thunderstorms makes it possible to expand our knowledge about meteorological phenomena, pollution, and air ionization. Long-term measurements of atmospheric electricity parameters can improve weather forecast models and enable the study of climate change and weather systems [1,2]. Electric charges on ions are responsible for the flow of electric current in the atmosphere. They attach to aerosol particles. As a result, air pollution can be monitored by measuring air conductivity, electric field, and current [3]. The study of the structure of lightning discharges and the transport of electric charges can be used to design lightning protection systems [4]. Moreover, monitoring the atmospheric electric field can provide valuable data for studying space weather phenomena such as geomagnetic storms, solar flares, and interactions between the solar wind and Earth’s magnetic field [5,6,7,8]. Understanding these processes is critical to space exploration, satellite communications, and navigation systems.

The issues presented in the paper, however, concern ground measurements of the electric flux through a near-surface, which is described by the general equation:(1)$$q=\varepsilon \oint_{S}^{}\vec{E}{}^{\circ}\hat{n}dS$$
where *q* is the accumulated electric charge, $\vec{E}$ is an electric field vector, $\hat{n}$ is the normal unit vector perpendicular to the observation surface *S*.

Maxwell current measurements in atmospheric electricity, whether during a thunderstorm or in fair weather, are usually made with a piece of metal placed horizontally to the ground (this sensor is commonly referred to as a flat antenna or a Wilson plate for fair weather measurements) [9,10,11] or a long wire antenna [12,13,14]. An antenna isolated from Earth’s surface constitutes an atmospheric capacitor.

Depending on the parameters of the external measuring circuit attached to the sensor, it can measure parameters such as:Derivate of the electric field *dE/dt*;Electric field vector *E*;The conduction current *I_cond_*;Displacement current *I_disp_*.

The external measurement circuit, which is usually capacitive, will have a current of *dQ_surf_*⁄*dt*, where *Q_surf_* is the charge on the measurement antenna, and *dt* is the observation time derivative. When there is a non-stationary electrical state in the atmosphere, the displacement current flows through the measuring circuit first and then the conduction current (e.g., the process of discharging the measuring capacitor). For quasi-stationary processes, the source of the displacement current density will be the change in space charge density over time, i.e., the change in the electric induction vector *D*.

In theoretical considerations of methods for measuring current in the atmosphere, Gauss and Ampere–Maxwell equations should be used. However, for a fixed sensor geometry, choosing appropriate sensor surfaces to integrate the electric charge on these surfaces or the road limiting this surface may prove difficult or impossible. A unique Gaussian surface should also be determined to calculate the electric field, and a specific Ampere loop should be used to determine the magnetic field and current density. A different approach was presented by Krider and Musser [9], who determined the Maxwell current (displacement component) in a thunderstorm environment from measurements at the ground using a flat antenna when the electric field concerning time equals zero. This method does not apply to measurements during fair weather conditions.

We suggest using a spatial analysis of the distribution of the electric field (*x*, *y*, *z*) in the presence of a measurement sensor (Figure 1) because the components of the electric field vector *E* significantly change their direction and value when in contact with the sensor [15,16].

## 2. Basic Rules

Due to electric induction, an electric charge is built up in the sensing plate measuring sensor depending on the rate of change of the electric flux perpendicular to the sensor surface. In practice, it can be described by the displacement current density vector *J_disp_*. At the same time, free ions represented by the conduction current density vector *J_cond_* flow along the force lines of field *E.* According to Maxwell’s equation, the displacement current density is equal to:(2)$$\vec{J_{disp}}=\varepsilon_{0}\frac{\vec{dE}}{dt}=\frac{\vec{dD}}{dt}$$

or current displacement value:(3)$$I_{disp}=\varepsilon_{0}\frac{d}{dt}\left(\oint_{S}^{}\vec{E}{}^{\circ}\hat{n}dS\right)$$

This current is not related to the physical movement of the charge but only to the change in the electric flux through the surface (a unique Gaussian surface perpendicular to the electric field) over time. 

When conduction current *J_cond_ =* λE, where λ—electric air conductivity (positive and negative) and displacement current *J_disp_* are combined, they have continuity properties, even though they are not continuous individually [17]. After that, we used a known integral form of the Ampere–Maxwell law:(4)$$\oint_{L}^{}\vec{B}d\vec{L}=\mu_{0}\left(I_{cond}+I_{disp}\right)$$
where *B* is the magnetic field flux vector, and *L* is the border of the observed surface.

In the space above the sensor, the current density, the strength of the electric field *E,* and electric induction *D* can be described by the relationships and laws of Gauss’s, Stok’s theorem, Ampere–Maxwell, Faraday’s law, and the principle of charge conservation. Selecting the appropriate space volume *VOL* and the area *S* stretched over this space is needed. The current densities in this space *J_cond_* and *J_disp_* can be written as equivalent currents on the sensor’s surface. The charge contained outside the *VOL* region can be replaced by a charge induced on the surface *S*. This charge will be the current source in the measurement system (Figure 2).

The described conversion of currents in the atmosphere to currents in the measurement circuit, due to the presence of two centers with different electrical conductivity, i.e., the atmosphere and the metal surface of the sensor, requires tracking the relationship between the vertical and horizontal components of the *E* fields at the border of the media. The component *E_z_* (in this case) at the boundary of the media experiences discontinuity, resulting in a charge accumulating on the sensor conductor’s surface. The electric field between two charged conducting plates is equal to:(5)$$\vec{E}=\left(\frac{\tau }{\varepsilon_{0}}\right){}^{\circ}\hat{n}$$
where τ is the charge density accumulated on the plates.

The electric boundary charge is generated on the sensor’s surface and can be measured in an external circuit.

## 3. Sensors Configuration Analysis

In the analysis presented in the paper, several assumptions were made, such as forcing the vertical component of the *E* field in the form of a constant value (Figure 3). Analyzed sensors were placed between upper and down electrodes. The electric potential of the upper equalizing layer (upper electrode) is *V_s_* = 1000 V (this value was used during static analysis). The down electrode was ground with electric potential *V_gnd_* = 0 V. Distance between them was defined as value *h_c_* = 10 m. It is associated with a typical mid-latitude fair weather atmospheric electric field value of 100 V/m. This electric field strength value is related to electric field strength on a fair weather day.

The inductive component representing the displacement current was determined by the rate of change of the electric flux relative to the plane perpendicular to the sensor. The sensors with different geometries adopted for the analysis were placed flush with the ground surface, and each time, they were directly grounded in an axisymmetrical point. The following three sensor geometries were taken into account:Flat plate sensor;The sensor in the shape of a grater;Ring-shaped sensor.

The models of the sensors described above were prepared with the ANSYS/Maxwell 3D simulation environment. The main observation parameters for each configuration were the electric field force lines and the sensor collection surface of the electric charge. These results were verified using the example of the charge accumulated between the upper and down electrodes. It has been described in the preliminary version of this research presented in the paper [18]. The latter parameter is necessary to determine the current density in the atmosphere based on measuring the current in the external circuit. The sensor was a homogeneous metal surface placed horizontally in most measurement applications.

The primary issue to consider was current in the external circuit if the sensor is oriented vertically, e.g., to maintain the symmetry of a ring shape, or will contain horizontal and vertical elements, e.g., a grater. Similar issues have been described in the literature, e.g., [17] in the past. Electric field distribution *E* (or potential *V*) parameters were considered in each case. The surface charge τ induced on the conductor (sensor) is equal to:(6)$$\tau =-\varepsilon_{0}\frac{dV}{dn}$$
where $\frac{dV}{dn}$ is the normal derivative of potential *V* at the surface. Analyses were performed for the electric simulation type, as excitation was defined as time-dependent changes.

### 3.1. Flat Plate Sensor

The simplest model taken under investigation was the model of the flat plate sensor. Its shape is presented in Figure 4. The main dimensions of the model were sensor diameter *d* = 1 m and depth of the hole *h* = 0.3 m.

The sensor was located at two positions: equal to ground level and at height *A* = 1 m above them. Figure 5 presents the exemplary electric field strength distribution in three dimensions *E(x,y,z)* for sensor positioning equally to ground level and at height *A* = 1 m above them. This distribution is reduced to a 2D cross-section defined as a Y-Z surface for better readability.

As might be expected, electric field strength vectors close to the conducting sensor deviate from vertical directions and lead to its surface. This aspect is described in the next point of the paper. The electric charge accumulated on the sensor equal to ground level had a value of *Q*_1_0_
*=* −7.31 × 10^−10^ [C]. However, if the sensor was positioned at height A = 1 m above ground level, this value was nearly four times higher *Q*_1_1_ = −2.42 × 10^−9^ [C].

### 3.2. Grater Shape Sensor

The second sensor model was the grater shape (Figure 6). Its main dimensions were: Sensor diameter *d* = 1 m;The height of the sensor *H* = 0.1 m;Depth of the ground hole *h* = 0.3 m.

Excitation assumed for simulations is described above as a static electric field with a strength equal to *E* = 100 V/m.

**Figure 6 sensors-23-09627-f006:**
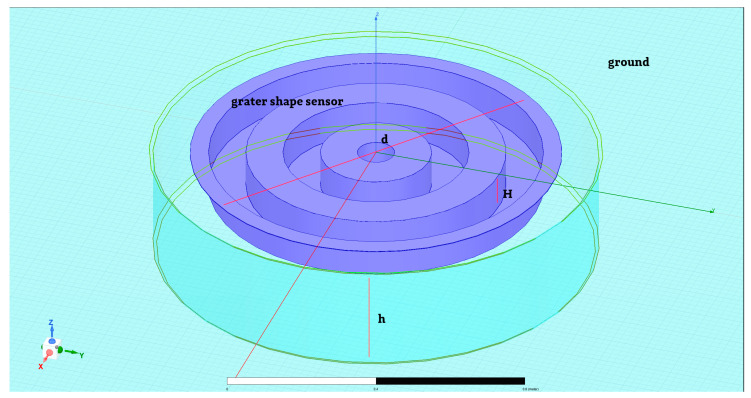
The grater shape sensor model was taken into consideration.

The sensor was located equal to ground level and at *A* = 1 m above them. Figure 7 shows the exemplary distribution of the electric field for given boundary conditions in three dimensions *E(x,y,z)* for the grater sensor placed equal to ground level and next at the height *A* = 1 m above them.

The electric charge that accumulated on the sensor equal to ground level had a value of Q_2_0_ = −5.51 × 10^−10^ [C], and in case the sensor was placed at height *A* = 1 m above ground level, it had a value of Q_2_1_ = −2.87 × 10^−9^ [C].

### 3.3. Ring-Shaped Sensor

The last sensor was the ring-shaped sensor (Figure 8). Its dimensions were: sensor diameter *d* = 1 m, the height of the sensor *H* = 0.3 m, and depth of the ground hole *h* = 0.3 m. Excitation for simulations was defined as a static electric field with a strength equal to *E* = 100 V/m.

The sensor was located equal to ground level and *A* = 1 m above them. Figure 9 shows the electric field distribution for given boundary conditions in three dimensions *E(x,y,z)* for flat plate sensors equal to ground level and *A* = 1 m above them.

The electric charge accumulated on the sensor at height *A* = 0 m above ground level had a value of *Q*_3*_*0_ = −1.27 × 10^−9^ [C]. The electric charge accumulated on the sensor at height *A* = 1 m above ground level had a value of *Q*_3_1_ = −3.92 × 10^−10^ [C].

For a sensor in the form of a ring and a grater, the surface density of the electric charge can be heterogeneous. Applying Gauss’s law to these shapes to determine the spatial charge based on the E-field is inappropriate since the surfaces in question are not so-called Gaussian surfaces. The distribution of the vector *E(x,y,z)* shows the space around the sensor representing the measured current in the external circuit. The *E_y_* component is presented for vertically oriented sensors since the E-field is perpendicular to the conductor surface (Figure 9).

## 4. The Collective Surface of the Sensors

The nature of the presented currents is volumetric and spatial. The current density vector describes them $\vec{J}$ (it contains all current components). The relation between the total current *I* through a surface and the current density $\vec{J}$ might be calculated on the geometry of the sensor:(7)$$I=\int_{S}^{}\vec{J}{}^{\circ}\hat{n}da$$

In this dependence, there is a component normal to the surface. It is imperative. The knowledge of the collecting area allows us to calculate the surface current value and assess the measured area’s practical size [19,20]. The shape of the selected collecting area is shown in Figure 10, Figure 11 and Figure 12. It has been calculated with the Matlab environment, which has imported results from Ansys. Next, vectors of electric field strength characterized by each calculation point were analyzed as percentage deviation to the vertical direction. If this deviation was more remarkable than the assumed 5% taken as a reference level, this point could influence the analyzed sensor. The results of vector decomposition for each analyzed sensor are shown in the following figures (Figure 10, Figure 11 and Figure 12). They present a 2-dimensional cut in the XZ surface (the Y coordinate was equal to zero)—calculation points were distributed with *a* = 10 mm distance.

In order to compare the parameters of the sensors, the term “effective area of the antenna” was used. The effective area is the horizontal cross sections above the antenna through which the electric field line ends on the sensors’ surface [11,17,20]. The air–earth current density can be determined based on the collected current in the external circuit of the sensor and the effective area. The effective area is approximately equal to the geometric area of the sensor when placed on the ground. In this case, the accumulated current is small, in the order of pA, and the measurement is carried out at the GEC electrode, which can cause significant distortion [10]. Figure 10 shows configurations of electric field lines obtained for the flat sensor (Wilson Antenna) model. For a sensor placed at ground level, as in Figure 10a, most field lines extend perpendicularly to the sensor’s surface. They are equal to the horizontal component—*E_z_*. When the flat sensor is located at a height of 1 m in the shaded areas, the component in the E field vector begins to dominate the *E_x_* component.

The effective area increases significantly, as seen in the vertical section of Figure 10b. At a height of 1 m, it is about 4.4 m. The sensor of the grater type has a similar size of the effective area (Figure 11) as the flat antenna. Vertical elements in the geometrical structure of the antenna do not have a significant impact on the shape of the effective area. Figure 12 shows the electric field of a ring-type sensor. The height of the ring, which is equal to 25 cm, causes the collecting surface in the vertical section to be about 2 m. Placing the sensor at a height of 1 m results in a collecting surface equal to almost 6 m. On this basis, it can be concluded that a sensor with only a vertical geometric configuration can have a large effective area. In this case, the dominant vector component of the electric field will be the *E_x_* component. On the collecting surface of the sensor, there may be only a vertical component of the electric field to this surface. Therefore, depending on the geometric configuration of the sensor (vertical or horizontal elements), the *E_x_* and *E_z_* components are coupled and change the value and direction. The shaded surfaces in the figures show the curving of the electric field’s force lines. The final value of the effective surfaces can be determined by approximating the lower areas of these surfaces with a curve (Figure 10b, Figure 11b and Figure 12b). The most exciting sensor was the sensor with a ring shape. Its sensitivity was more significant than others. In horizontal and vertical directions, it was over 2 m. It means this type of sensor collects more electric charges than others, so measuring results like sensor current or voltage might differ and should be considered during results post-processing.

## 5. Time Response to Slow Changes in the Electric Field

The sensors’ response to the electric field’s time change allows us to determine their basic dynamic parameters, i.e., sensitivity to the displacement current [12,21]. The change in electric flux over time equals the change in electric charge on the sensor, which is nothing more than the definition of displacement current. The presented simulations are intended to compare sensors for detecting displacement currents. Excitation was defined as a very slow-changing slope, as presented in Figure 13.

Its value changed from *U*_0_
*=* 0 V to *U*_1_ = 1000 V in time *t_s_* = 1 s. Because the time dependency simulation type has to be changed to transient, simulation time was set to *t_sim_* = 1.2 s and time step as *t_step_* = 0.01 s. This current is described as the current in the external circuit. A sensor placed near the ground represents a simpler RLC circuit, like in Figure 14, so time response is a significant factor in describing its functionality. Resistance *R* results in the sensor ground connection, capacitance *C* is the sensor ground result, and inductance *L* is determined by sensor construction and geometry. The observed quantity was current density flows by thin conductor between sensor and ground and next integrated by its cross-section surface area. Figure 15 shows the variation in the electric current flows by sensor grounding connected and integrated by its cross-section surface area. 

The sensor’s response to this excitation is in the form of fluctuations on the slopes, associated with the transient state (can be approximated by the envelope) and a flat section (steady state). The simulated variation in the current depends on the electrical capacitance of the sensor and the electric charge induced on its surface and can represent the displacement current. As shown in the Figure 15 reaction, each sensor on slowly changing electric field strength has high-frequency oscillations. The oscillations result from the external circuit (RLC) parameters through which the electric current flows. They occur at the beginning and end of the excitation slope. The oscillations result from the parameters of the external circuit through which the electric current flows. The ring-type sensor’s current value and transient response are the highest (Figure 15c). The higher current value in the steady state is for all sensors placed at the height of 1 m (green color). This effect exists due to the larger effective area of the sensor. The transient response of the sensor is independent of the height of the sensor. Assuming that the derivative of the electric field concerning time or the second derivative of the electric potential represents the density of the displacement current [17]. Figure 15c shows the highest value of this current in a steady state in the ring-type sensor.

## 6. Discussion and Conclusions

The paper describes the characteristics of selected measurement sensors of various geometric shapes that measure the atmosphere’s electricity. This type of sensor analysis has not yet been published. Based on the presented analysis, it is impossible to state clearly, based on voltage or current measurements in an external electrical circuit, a significant relationship between the shape of the sensor and the physical quantity in the atmosphere.

Due to the inhomogeneous surface structure, the grater sensor has a spatial distribution of the potential with a significant value accumulated around the sensor. The coexistence of many small vertical and horizontal sensor structures shapes its characteristics.

It can be challenging to analyze the conduction and displacement density vector in connection with electrical phenomena in the atmosphere (e.g., fair-weather conditions or thunderstorms) based on the measurement of the current in the external circuit attached to the sensor. The movement of the current-forming carriers to the measuring sensor may result in a conduction or displacement current or simultaneously these currents (depending on the time). The currents may pass one way or another or exist independently [12,21].

The surface of the sensors does not differ significantly from their geometric sizes. The shape of the sensor may not be a significant element in distinguishing individual currents in the atmosphere based on the current in the external circuit, which will be the subject of further model calculations and experiments [22].

The characteristics of sensors for time-varying electric fields and the impulse response also have been taken for investigation. The obtained results confirm static analysis results and get information about the sensor’s sensitivity regarding electric field strength slow changes in time. Such an analysis may indicate the suitability of the selected sensor to identify the displacement current in the atmosphere better.

The measurement of the displacement current in the atmosphere is possible but exceedingly tricky, using the measurement of the magnetic component *B* and the selected surface surrounded by the Amperian loop. This fundamental method is based on the Ampere–Maxwell law [23,24]. A preliminary study of such a method is also under investigation.

The Maxwell current components may be proportional to the horizontal cross-sectional area (e.g., conduction current) and surface area (e.g., displacement current) of the measuring sensor [22]. This observation is related to the distribution of electric field force lines on a conductor placed in an electric field. However, the normal component of the electric field to the sensor surface plays a significant role. We suggest using a spatial analysis of the electric field distribution (*Ex*, *Ey*, *Ez*) in the presence of a measurement sensor. We suggest using a spatial analysis of the distribution of the electric field (*x*, *y*, *z*) in the presence of a measurement sensor because the components of the electric field vector E significantly change their direction and value when in contact with the sensor. The current at the surface may be measured directly utilizing an insulated sensor set level with the ground surface. However, due to the measurement of the air–earth current at the GEC electrode (i.e., at Earth’s surface), the current may not be representative. For this situation, it was proposed to raise the sensor above the ground and calculate the 3D distribution of the electric field.

## 7. Summary

In summary, our analysis leads to the following conclusions:The method showed slight differences in detecting air–earth current components for commonly used sensors. In our opinion, simulations should be performed for other complex geometric shapes or a sensor composed of elements of different shapes.3D simulations of the electric field in the presence of sensors should be used.Adjusting the geometry of the measurement sensor to the application of measuring electric fields and currents in the atmosphere can improve the detection of individual components of the GEC structure. However, this requires further research.Measurements of electric currents in the atmosphere are difficult due to various transport of electric charges, inductive effects caused by moving electric charges, free and bound charges, and very low amplitudes. Therefore, the development of new sensors is necessary.

## Figures and Tables

**Figure 1 sensors-23-09627-f001:**
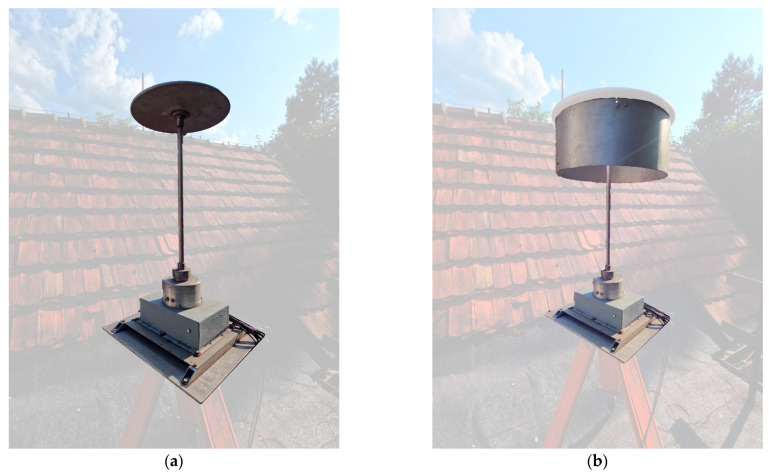
Exemplary pictures of analyzed sensors: (**a**) flat shape, (**b**) ring shape.

**Figure 2 sensors-23-09627-f002:**
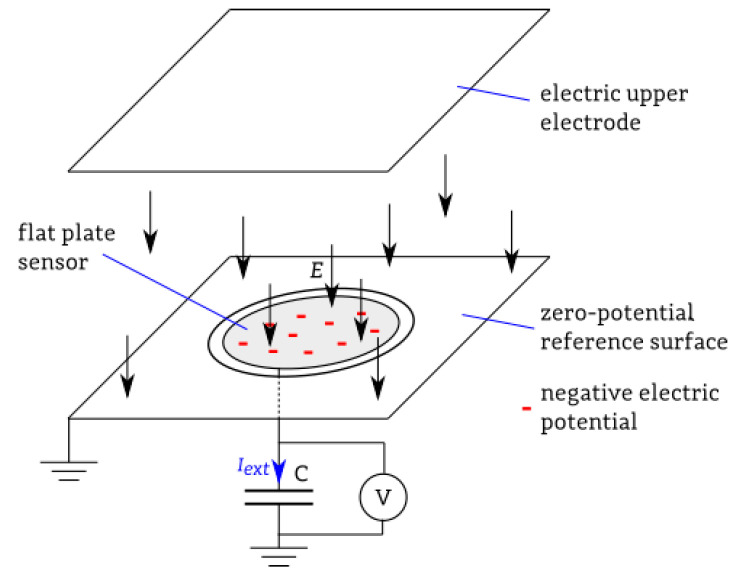
Exemplary view of field plate sensor and measured current (I_ext_—current in external measurement system). Arrows show electric field direction.

**Figure 3 sensors-23-09627-f003:**
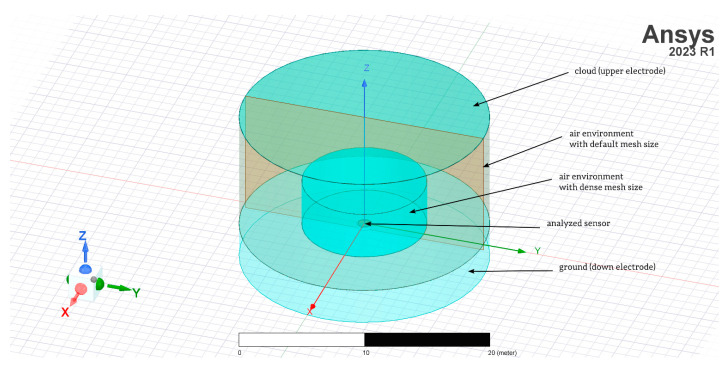
Exemplary view on flat plate model assumptions.

**Figure 4 sensors-23-09627-f004:**
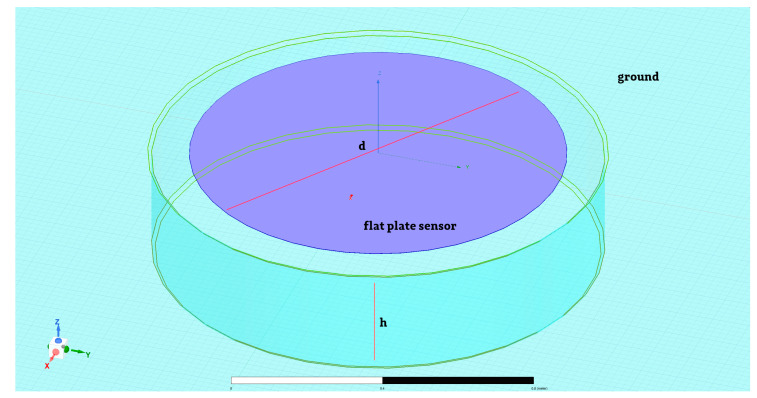
The flat plate sensor model is taken into consideration.

**Figure 5 sensors-23-09627-f005:**
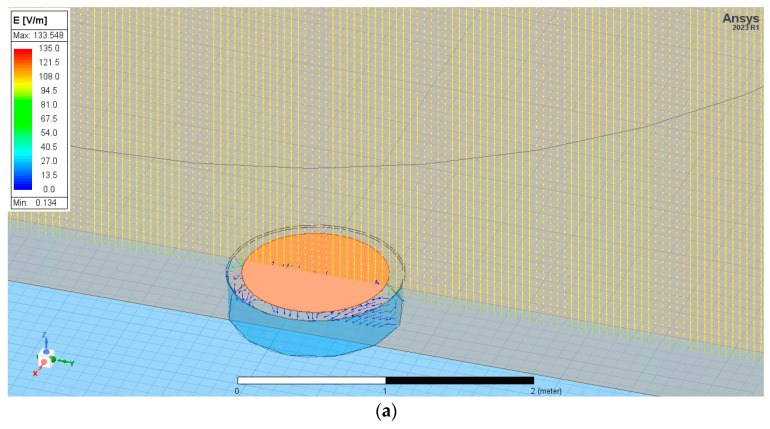
The electric field distribution near the flat plate sensor is placed equal to ground level (**a**) and at height A = 1 m above ground level (**b**).

**Figure 7 sensors-23-09627-f007:**
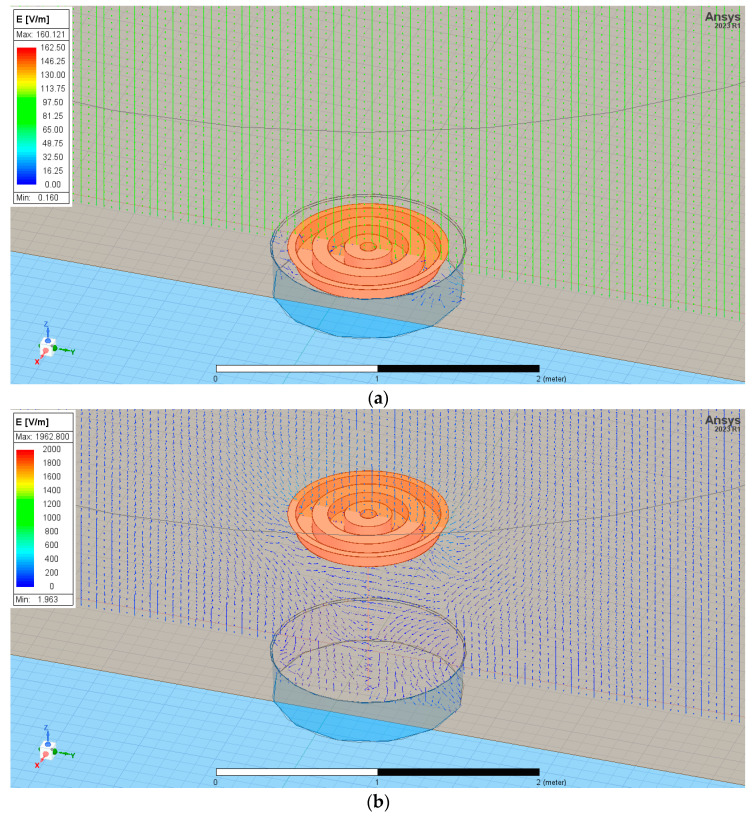
The electric field distribution near the grater-shaped sensor placed equal to ground level (**a**) and *A* = 1 m above ground level (**b**).

**Figure 8 sensors-23-09627-f008:**
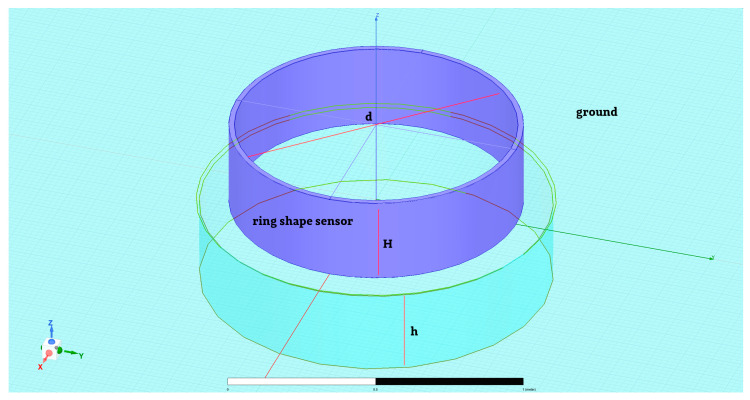
The ring-shaped sensor model was taken into consideration.

**Figure 9 sensors-23-09627-f009:**
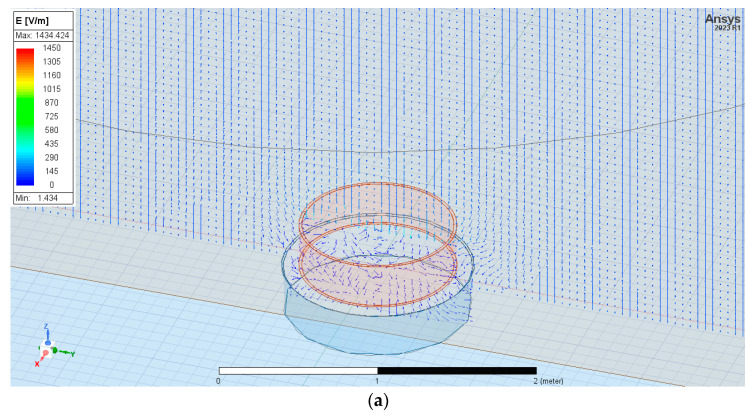
The electric field distribution near the ring-shaped sensor placed equal to ground level (**a**) and *A* = 1 m above ground level (**b**).

**Figure 10 sensors-23-09627-f010:**
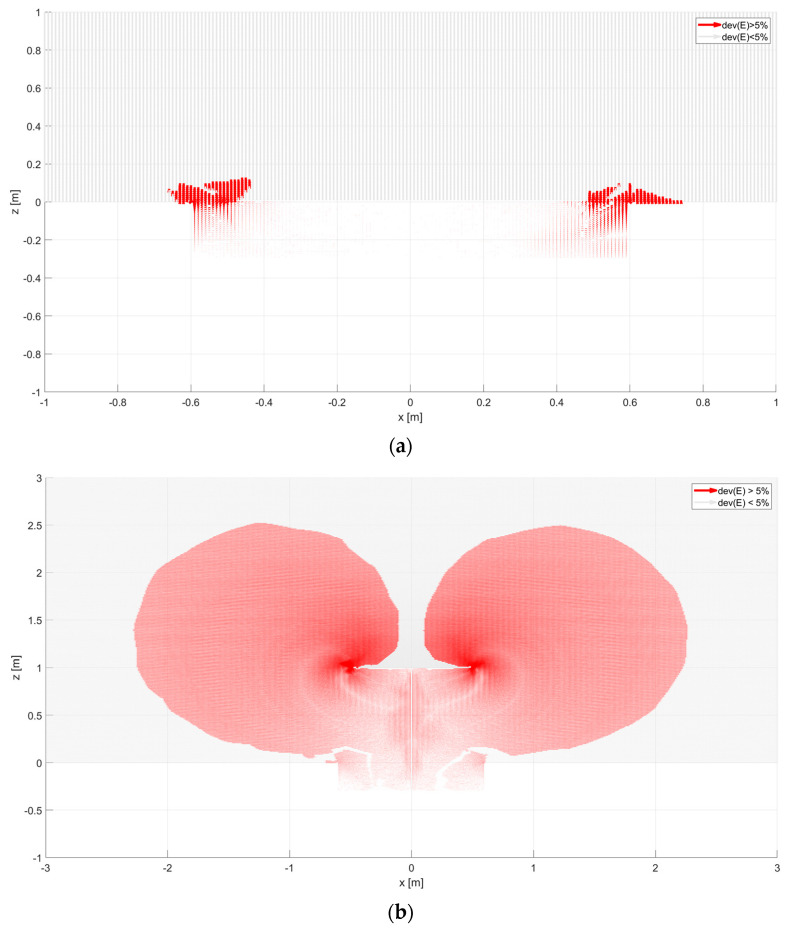
Deviation of electric field vector for flat type sensor: (**a**) placed equal to ground level (*A* = 0 m), (**b**) placed above ground level (*A* = 1 m). Areas shown in red represent electric field lines that will reach the sensor surface, reducing the field component. These areas represent the directional characteristics of the sensor. In the bright areas above the sensor, the electric field lines arrive perpendicularly, i.e., without changing their components.

**Figure 11 sensors-23-09627-f011:**
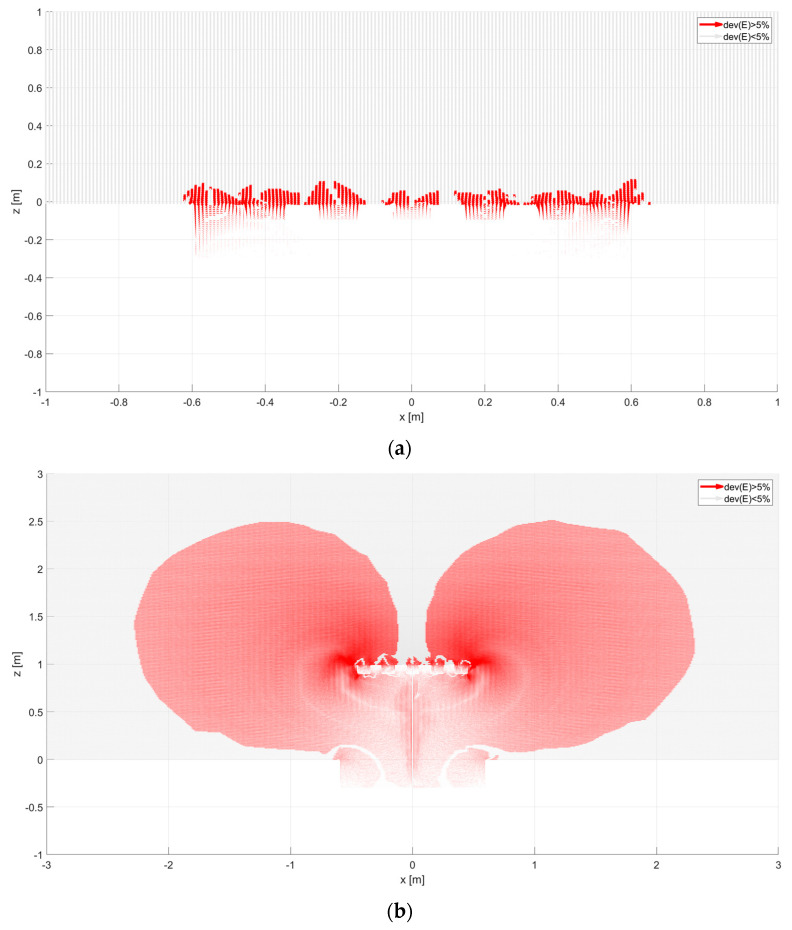
Deviation of electric field vector for grater type sensor: (**a**) placed equal to ground level (*A* = 0 m), (**b**) placed above ground level (*A* = 1 m). Areas shown in red represent electric field lines that will reach the sensor surface, reducing the field component. These areas represent the directional characteristics of the sensor. In the bright areas above the sensor, the electric field lines arrive perpendicularly, i.e., without changing their components.

**Figure 12 sensors-23-09627-f012:**
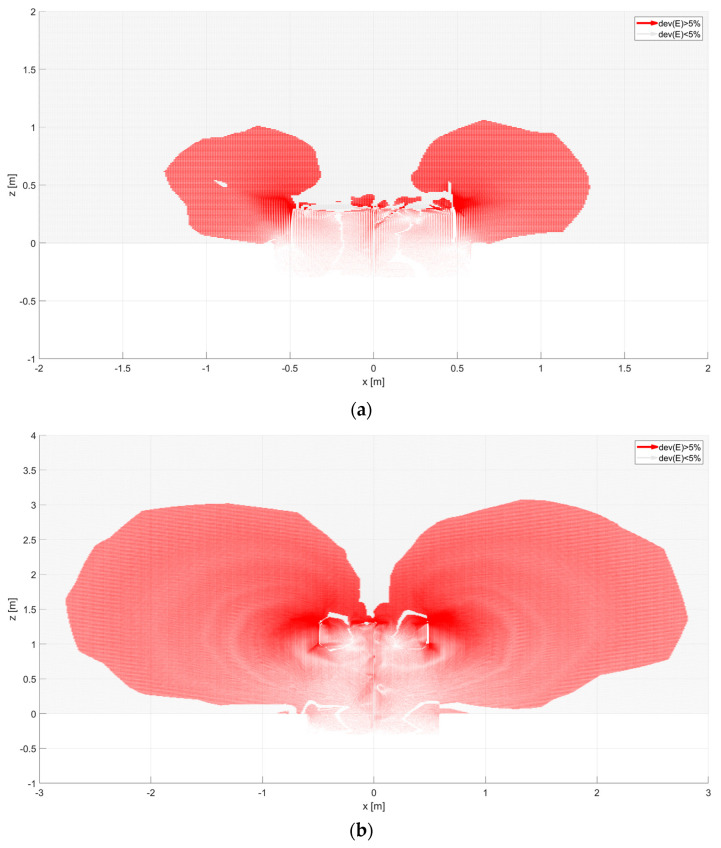
Deviation of electric field vector for ring type sensor: (**a**) placed equal to ground level (*A* = 0 m), (**b**) placed above ground level (*A* = 1 m). Areas shown in red represent electric field lines that will reach the sensor surface, reducing the field component. These areas represent the directional characteristics of the sensor. In the bright areas above the sensor, the electric field lines arrive perpendicularly, i.e., without changing their components.

**Figure 13 sensors-23-09627-f013:**
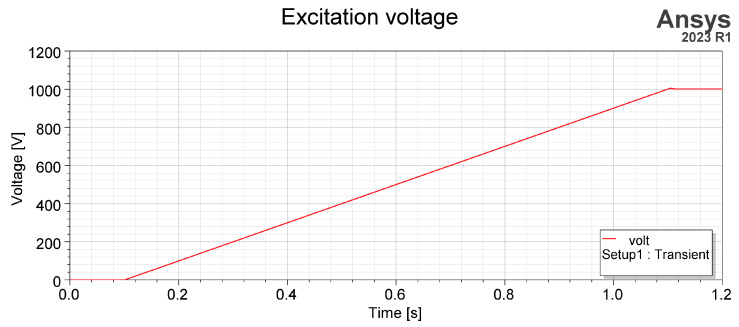
Time variable excitation—1000 V/s rising slope.

**Figure 14 sensors-23-09627-f014:**
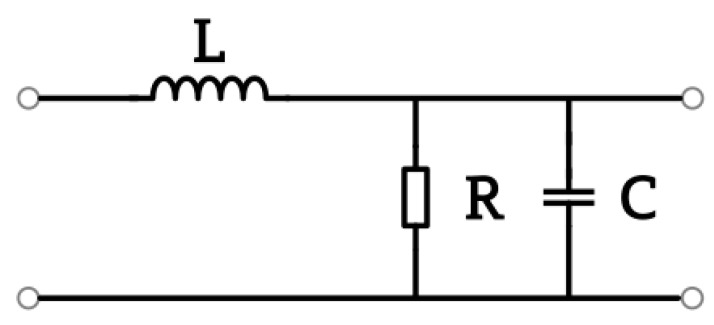
Simple electrical diagram describing idea time response simulations.

**Figure 15 sensors-23-09627-f015:**
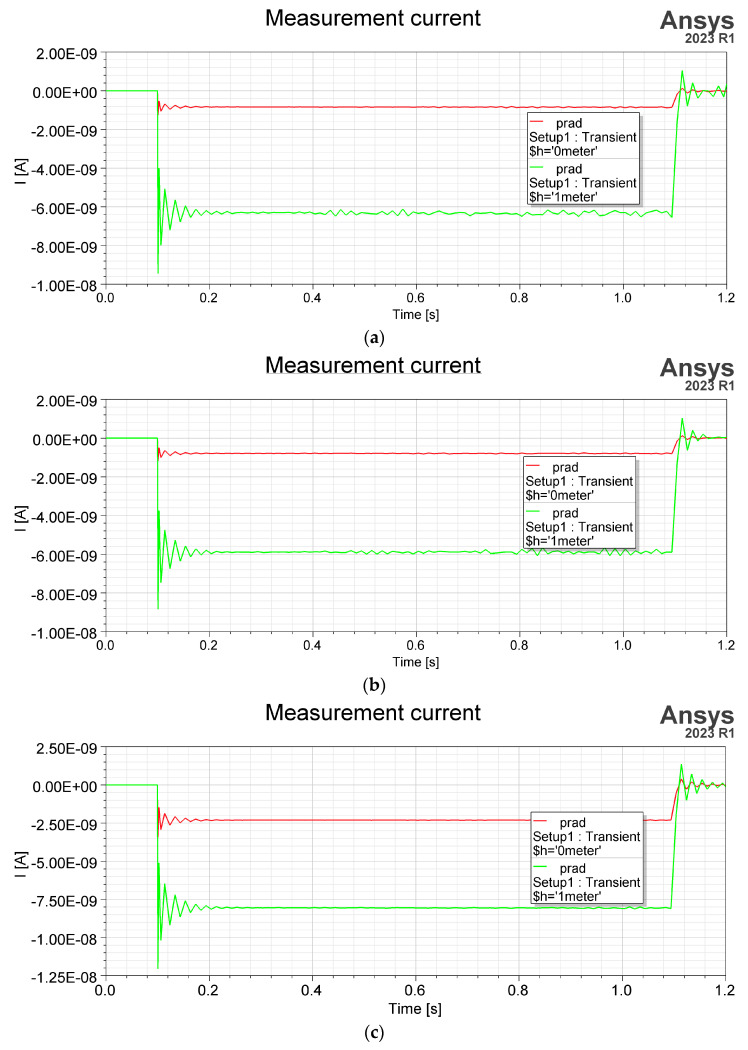
Respond to excitation: (**a**) flat shape sensor, (**b**) grater shape sensor, (**c**) ring shape sensor, placed equal to ground level (red color) and at the height of 1 m (green color).

## Data Availability

Data are contained within the article.

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
