# Peer review of "Simulation Analysis of Sensors with Different Geometry Used in Measurements of Atmospheric Electricity†"

_sensors, 2023, doi:10.3390/s23249627_

Round 1

Reviewer 1 Report

Comments and Suggestions for Authors

The paper needs the major revision as well as the extensive correction of English, see attachment docx file.

Comments on the Quality of English Language

Please check the English carefully, as well as the terminology.

Author Response

We are very grateful for the reviewer's constructive comments. We provide detailed responses in attached file (marked in italics).

Reviewer 2 Report

Comments and Suggestions for Authors

The work is devoted to studying the effectiveness of current collectors of various forms when measuring vertical current in the atmosphere. First, the theoretical aspects of this problem are considered, and then the results of calculations in the ANSIS system are presented. From a mathematical and physical point of view, there are no complaints about the material presented, however, it seems to me that the issues raised in the work do not fully illuminate the problem under study. Three types of the most “popular” current collectors are considered, but there are no conclusions about their advantages and disadvantages. This is especially true for a “grater”-shaped collector, which is used, as a rule, to separate the conduction current and the Maxwellian displacement current. It seems to me that Chalmers' works on the measurement of atmospheric currents under various conditions should be added to the bibliography. This work does not address many issues related to the operation of the current collectors under consideration in real atmospheric conditions, such as precipitation, snowfall, etc. The electrode effect, which significantly affects the operation of current measuring systems, is also beyond the scope of consideration. The illustrations in the work look unclear, it is difficult to make out the direction of the electric field vector near metal surfaces. At the same time, the work is of undoubted importance in terms of methods for calculating current collectors using modern electrodynamics modeling systems and can be useful to a wide range of specialists, especially beginners, in this field of research.

Author Response

We are very grateful for the reviewer’s constructive comments. We provide detailed responses below (in italics).

1. The influence of the electrode effect has been expanded.

 2. So, we confirm this is a significant problem in measuring electric current in the atmosphere.

 3. The figs. have been corrected to consider the electric field vectors. Our idea was to show the characteristics of the sensor collecting surfaces. Therefore, the selected spatial domain did not provide details of the electric field vectors.

Round 2

Reviewer 1 Report

Comments and Suggestions for Authors

I have got the answers to my questions. The quality of figures has been essentially improved and the necessary explanations are available now.

Comments on the Quality of English Language Only I may recommend to show the text to the native speaker to correct English.